# Potential Mechanisms of Bisphenol A (BPA) Contributing to Human Disease

**DOI:** 10.3390/ijms21165761

**Published:** 2020-08-11

**Authors:** Ilaria Cimmino, Francesca Fiory, Giuseppe Perruolo, Claudia Miele, Francesco Beguinot, Pietro Formisano, Francesco Oriente

**Affiliations:** Department of Translational Medicine, Federico II University of Naples and URT “Genomic of Diabetes” of Institute of Experimental Endocrinology and Oncology, National Council of Research (CNR), 80131 Naples, Italy; ilaria.cimmino@unina.it (I.C.); francesca.fiory@unina.it (F.F.); giuseppe.perruolo@unina.it (G.P.); c.miele@ieos.cnr.it (C.M.); beguino@unina.it (F.B.); foriente@unina.it (F.O.)

**Keywords:** bisphenol A, receptors, transcription factors, epigenetics, metabolism, cancer

## Abstract

Bisphenol A (BPA) is an organic synthetic compound serving as a monomer to produce polycarbonate plastic, widely used in the packaging for food and drinks, medical devices, thermal paper, and dental materials. BPA can contaminate food, beverage, air, and soil. It accumulates in several human tissues and organs and is potentially harmful to human health through different molecular mechanisms. Due to its hormone-like properties, BPA may bind to estrogen receptors, thereby affecting both body weight and tumorigenesis. BPA may also affect metabolism and cancer progression, by interacting with GPR30, and may impair male reproductive function, by binding to androgen receptors. Several transcription factors, including PPARγ, C/EBP, Nrf2, HOX, and HAND2, are involved in BPA action on fat and liver homeostasis, the cardiovascular system, and cancer. Finally, epigenetic changes, such as DNA methylation, histones modification, and changes in microRNAs expression contribute to BPA pathological effects. This review aims to provide an extensive and comprehensive analysis of the most recent evidence about the potential mechanisms by which BPA affects human health.

## 1. Introduction

Persistent organic pollutants (POPs) are organic compounds resistant to degradation and are able to bioaccumulate in the environment, affecting human health [1]. Today, as in the past, many POPs are used to produce fertilizers, pharmaceuticals, and pesticides. As a consequence, these chemicals have contaminated water, air, and soil, and high concentrations of POPs have been found in animal and human tissues, milk, and blood [2,3,4]. Bisphenol A (BPA) is an organic synthetic compound with a molecular weight of 228 Da and the chemical formula (CH_3_)_2_C(C_6_H_4_OH)_2_. It is included in the group of diphenylmethane derivatives and bisphenols, with two hydroxyphenyl groups [5,6]. This chemical compound was firstly synthesized in 1891, by the Russian chemist Aleksandr P. Dianin, who combined phenol with acetone in the presence of an acid catalyst. In the 1950s, scientists discovered that the reaction of BPA with phosgene (carbonyl chloride) produced a clear hard resin known as polycarbonate, which became widely used in the packaging for food and drinks, safety and medical devices, thermal paper, and dental compounds [7,8,9,10,11,12,13,14,15,16]. BPA half-life is about 4.5 days in water and soil, while is less than one day in the air, because of the low volatility [17,18]. However, BPA presence in the air is due to the attachment to the solid particulates present in the atmosphere. Thus, the inclusion of BPA in the POPs category is controversial. Indeed, although not technically a persistent organic pollutant because of its short half-life, it is often grouped together with other POPs, as it can accumulate in human tissues and organs, and contribute to the pathogenesis of several diseases [6,19,20,21]. The first evidence for the mechanisms of action of BPA was obtained in 1936 by Dowds and Lawson who discovered its estrogenic properties in vivo [22]. In 1997, the involvement of estrogen receptors, ERα and β, in BPA action was described, while other mechanisms emerged later [23,24].

Several routes of exposure to BPA have been described, including the digestive system (ingestion), the vertical transmission (maternofetal), the respiratory system (inhalation), and the integumentary system (skin and eye contact) (Figure 1). BPA can be directly or indirectly released into the environment at any level of the life cycle of the product: production, consumption, or disposal [25].

This compound can be found as colorless crystals or as powder and can be released by plastic products into foods and drinks as a result of heating and acid or basic conditions. Indeed, exposure of polycarbonate plastics to high temperatures, for example by heating food stored in packages or baby bottles, increases the rate of BPA transfer to human body. In addition, contact with acid or basic compounds and the presence of high levels of sodium chloride or vegetable oils cause an increase in the release of BPA from polymeric materials [26,27].

BPA is able to cross the placental barrier and has been detected in human maternal and fetal serum and the human placenta. Thus, BPA can find its way into tissues and fluids in the human womb [28,29,30]. Furthermore, BPA can be also absorbed by inhalation or by contact. For example, the thermal paper of the receipts may release this compound through contact with the epidermis [12]. Moreover, very high plasma and urine levels of BPA have been found in the cashiers, the latter being more in contact with the thermal paper [11,13,14,15,16]. Other routes of exposure are the discharges of municipal wastewater treatment plants, the combustion of domestic waste, and the degradation of plastic materials [21].

Recent metabolic and toxicokinetic studies have shown a rapid oral absorption of BPA. Once absorbed, this compound is conjugated in the liver with glucuronic acid. BPA glucuronate is sufficiently stable and represents a valid exposure biomarker [31]. Although some controversial evidence indicates that BPA is not toxic to human health [25,32], several recent studies highlight its harmful effects. Because of its lipophilic nature (logP of 3.4), BPA has the ability to accumulate in different human and animal tissues, compromising their physiological functions and exerting deleterious effects on health [21,33,34]. Indeed, studies performed in humans, rodents, and cellular cultures suggest that this compound may be obesogenic through different mechanisms. By modulating PPARs, BPA induces adipogenesis, stimulates lipid accumulation in adipose tissue and liver, and perturbs cytokines levels. Furthermore, data obtained in human and different cell lines show that BPA interferes with thyroid hormones synthesis, secretion, and signaling. Due to its anti-androgenic action, BPA works as an agonist on estrogen receptors and antagonist on androgen receptors [35]. Recently, it has been shown that BPA interferes with spermatogenesis and impairs male reproductive function. In parallel, sperm motility is negatively affected by BPA in human, mouse, bovine, chicken, and fish [36]. BPA exposure has been also associated with an increased risk for hypertension and cardiovascular disease in humans and rodents, although the mechanisms are still unclear [37] (Figure 1). Interestingly, BPA affects glucose metabolism, onset and progression of several tumors, and immune function by binding different receptors, modulating transcription factors, and inducing epigenetic changes [38,39]. Most of these results have been obtained in humans, rodents, and cellular cultures. The public concern about the potentially harmful health effects of BPA resulted in a ban on many plastic products, particularly those used for infants and young children [40].

In this review, we will discuss the main molecular mechanisms by which BPA mediates its deleterious effects.

## 2. BPA Interaction with Specific Receptors

BPA belongs to the endocrine-disrupting class of compounds and exhibits hormone-like properties. Low doses of this compound induce adverse effects on reproduction and regulation of the immune system, hormone-dependent cancers, and metabolism [41]. Both in vitro and in vivo data have shown that BPA can bind several nuclear receptors, such as estrogen receptors (ERα and β), GPR30, androgen receptor (AR), thyroid hormone receptors (TRα and β), estrogen-related receptor gamma (ERRγ) and glucocorticoid receptor (GR) [23,41,42,43]. All these receptors may contribute to the adverse effect of BPA in human diseases.

### 2.1. Estrogen Receptors

Estrogens are involved in different physiological processes, including growth, development, and homeostasis of several tissues, through the binding and the activation of classical estrogen receptors, ERα and ERβ. These molecules are encoded by two separate genes located on human chromosome 6 and 14, respectively [44,45]. Besides estrogens, ERα and ERβ can bind a wide range of compounds with different structures, including BPA, which exhibit different binding preferences and relative binding affinity for both ER subtypes and ERs of different species [46,47].

BPA acts like estradiol, stimulating different cell responses, although its affinity for the estrogen receptor is lower and its activity is approximately 10,000 to 100,000 times weaker compared to the natural hormone 17 beta estradiol (E2) [46,48]. Indeed, Delfosse et al. investigated the interaction between BPA and ERα, demonstrating that this compound binds ERα through 42 van der Waals interactions, instead of the 51 involved in E2-ERα binding [6]. Another key point is the concentration of BPA able to exert significant effects. Surprisingly, BPA features a stronger estrogen-like activity at nanomolar doses than at micromolar doses [49,50,51].

In vitro studies have demonstrated similarities between the action of estrogen and BPA on the gene expression of adipogenic transcription factors [52]. In addition, both BPA and E2 have been reported to inhibit adiponectin secretion from human adipocytes in a non-monotonic dose-dependent manner [53]. BPA may affect body weight, too. Indeed Rubin et al. reported sex- and dose-dependent body weight differences in mice after early postnatal exposure to endocrine disruptors. Tissue-specific alterations in ER expression may further modulate the BPA effect on body weight [54].

BPA binding to estrogen receptors plays an important role also in tumorigenesis. In particular, BPA-ER interaction increases proliferation and migration of several ovarian cancer cell lines through a pathway involving Stat3 and ERK1/2 [55]. A wide variety of studies demonstrated that nanomolar doses of BPA significantly increase the proliferation of ER-positive and ER-negative breast cancer cells [56]. Moreover, Dairekee and co-workers reported that BPA inhibits the pro-apoptotic effects of the rapamycin suppressing signaling pathway mediated by p53 and BAX in human breast epithelial cells [57].

### 2.2. GPR30

In contrast to nuclear receptors genomic signaling, it has been recently proposed that the adverse effects of low dose BPA on human health could be mediated by membrane receptors in a non-genomic way in order to produce fast biological responses on specific cellular targets. In particular, the signaling pathway that involves GPR30, a non-classical ER, plays a key role in the deleterious effects of low dose BPA [50,58,59].

GPR30 is a seven-transmembrane domain receptor, firstly identified as an orphan member of the G-protein coupled receptor family in the late 1990s [42,60,61,62]. GPR30 mRNA is expressed in several tissues (e.g., placenta, lung, liver, prostate, ovary, placenta, and endothelium), with different expression patterns [42,61,62]. GPR30 mediates some rapid biological events elicited by E2 through the activation of different pathways, including generation of the second messengers Ca^2+^, cAMP, and NO, as well as activation of tyrosine kinase receptors, such as EGFR and IGF-1R, and induction of kinases like PI 3-kinase, PKB, and ERK family members [63,64,65,66,67,68,69].

Interestingly, Revnkar et al. have demonstrated that E2 affinity to GPR30 is 10-fold lower than ERα, while BPA affinity to GPR30 is about 50-fold higher than ERα [70,71].

The role of GPR30 in BPA-mediated detrimental effects on metabolism has been clearly demonstrated. Indeed, Wang et al. have indicated that GPR30 knockout (GPRKO) female mice are protected from high-fat diet (HFD)-induced obesity, blood glucose intolerance, and insulin resistance [72]. In parallel, Garcia-Arevalo et al. shed light on BPA interference with glucose metabolism, showing that BPA exposure causes impaired glucose tolerance, body weight gain, and reduced insulin secretion in mice [73,74]. We have recently demonstrated that low dose BPA increases GPR30 and the production of specific inflammatory proteins, including IL8, IL6, and MCP1α, both in cultured mature adipocytes and in stromal-vascular fraction cells isolated from mammary human adipose tissue biopsies [26].

GPR30 is widely expressed in different cell types and cancer cell lines and is overexpressed in endometrial, breast, and ovarian cancers [75,76]. Dong and collaborators have demonstrated that BPA, through GPR30, increases ERK1/2 phosphorylation and triggers a rapid biological response in both ER-positive and negative breast cancer cells [77]. In a mouse spermatocyte-derived cell line, GC-2 cells, low doses of BPA bind to GPR30 and activate the EGFR-MAPK pathway, with consequent activation of the c-Fos gene and inhibition of cell-cycle gene Cyclin D1 [78]. In males, GPR30 has been found to be particularly overexpressed in human seminoma tumors, the most frequent testicular germ cell tumor. Interestingly, the BPA-GPR30 complex induces testicular seminoma cell proliferation in vitro, and incubation with G15, a GPR30 antagonist, reverts this effect [79].

### 2.3. Androgen Receptor

Growing evidence supports the anti-androgen effect of BPA [80,81,82]. BPA is able to compete with 5α-dihydrotestosterone (DHT) for binding to androgen receptors (ARs). Several in silico studies have reported the ability of BPA to bind multiple sites on the AR surface through hydrophobic interactions [83,84]. The BPA-AR pathway is associated with adverse effects on spermatogenesis, steroidogenesis, atrophy of the testes, and alteration of adult sperm parameters, such as sperm count, motility, and density both in experimental animals and in humans [43,85]. These findings provide evidence that BPA induces several defects in the embryo, during postnatal and pubertal periods and adulthood. Indeed, this compound affects the hypothalamic-pituitary-testicular function by modulating androgen and estrogen synthesis as well as expression and activity of the respective receptors. The anti-androgenic effects on male reproductive function may be mediated by different mechanisms that involve receptor stabilization, dissociation of the heat shock protein 90, and nuclear translocation [86]. BPA’s ability to impair male reproductive function in humans has been evidenced by epidemiological studies. Li et al. demonstrated that men exposed daily to BPA show lower sexual function such as erectile and orgasmic function, sexual desire, reduced libido, and erectile ejaculatory difficulties compared to controls [87]. These defects are paralleled by higher BPA levels in urine and plasma samples [88].

### 2.4. Other Receptor Targets of BPA Action

The estrogen-related receptors (ERRs) belong to a family of orphan nuclear receptors that includes (ERRα, β, and γ) [89]. Although these receptors share a relevant homology with ER, they do not directly bind estradiol. Differently, BPA can interact with these receptors, despite its estrogen-like activity. In particular, several studies demonstrated a strong affinity of ERRγ to BPA even at nanomolar concentrations [90,91,92]. ERRγ is constitutively active and owns a ligand-independent transcriptional activity [93]. However, Zhang and co-workers have demonstrated that low doses BPA could trigger the expression of the MMP2-mediated pathway and the invasion of triple-negative breast cancer through ERRγ [94]. In addition, the silencing of ERRγ attenuated BPA-induced proliferation of breast cancer cells [56].

It has been demonstrated that BPA can interact with the glucocorticoid receptor (GR) with lower affinity, compared to cortisol or dexamethasone. According to Atlas et al., BPA could not be considered a full GR agonist but has a synergistic effect on adipogenesis [95].

Interestingly, human urinary BPA levels have been associated with higher T3 and lower TSH circulating levels [96]. Thus, since BPA displays structural similarities with T3 [97], the interaction between BPA and the thyroid hormone receptor (TR) has been investigated. In particular, BPA has been shown to bind TR, exerting both agonist and antagonist effects, and to directly affect thyroid function by increasing the expression of several genes involved in thyroid cell proliferation and activity [98]. Anyway, more data are needed to further clarify the effect of BPA exposure on the thyroid hormones’ pathway.

## 3. BPA Regulation of Transcription Factors

Growing evidence has shown the involvement of several transcription factors (TFs) in BPA action. In particular, some experimental evidence shows that induction of adipogenic TFs, such as PPARγ, C/EBPs, and Nrf2, plays a key role in the BPA “obesogenic effect”. Other studies suggest an important role of HOX family members and HAND2 protein in BPA-mediated detrimental effects.

### 3.1. PPARγ

Peroxisome proliferator-activated receptors (PPARs) are members of the nuclear receptor superfamily with a wide range of biological effects on metabolism, whole-body energy homeostasis, adipogenesis, cellular proliferation, differentiation, and immune response. This receptor family is comprised of three different subtypes (PPARα, β/δ, γ), all of which are important regulators of lipid and glucose metabolism in many different tissues including skeletal muscle, liver, adipose tissue, and gut. PPARγ activity is governed by the binding of small lipophilic ligands, mainly fatty acids, derived from nutrition or metabolism. Several studies suggest that BPA can modulate adipogenesis by inducing PPARγ, although the underlying molecular mechanisms are still unclear. Somm et al. indicate that both male and female pups prenatally exposed to low dose BPA (70 μg/kg/day) are overweight. However, at weaning, after postnatal BPA exposure via milk during lactation, only females show an increase in body weight, and this effect is associated with adipocyte hypertrophy and overexpression of proadipogenic transcription factors, such as PPARγ [99]. These results underline the importance of gender in PPARγ induction by BPA. In agreement, gestational BPA exposure enhances PPARγ expression in preadipocytes isolated from female, but not from male, sheep progeny [100].

Interestingly, PPARγ mediates BPA effects not only in adipose tissue but also in the liver. García-Arevalo et al. described that a subcutaneous injection of 10 µg/kg/day of BPA in mice upregulates the PPARγ gene in the liver and causes fasting hyperglycemia, glucose intolerance, and high levels of non-esterified fatty acids [73]. Similarly, Biasotto et al. indicated that administration of 5 μg/kg increases total body weight, fat mass, and hepatic PPARγ expression [101]. On the contrary, several conflicting results about the effect of BPA on PPARγ have been obtained in cellular cultures. According to Ariemma et al. and Biasotto et al. both low (0.1–1 nM) and high (80 μM) doses of BPA increased PPARγ in murine 3T3-L1 cells [101,102]. In contrast, Atlas et al. noted no differences in PPARγ 1 and PPARγ 2 expression in the same cells in response to BPA exposure [95].

In human cells, evidence in unclear. In adult human preadipocytes and in freshly cultured omental adipose tissue from children donors, BPA significantly increases PPARγ expression when used at different concentrations [103,104]. In contrast, PPARγ does not emerge as an essential mediator of BPA action in human adipose-derived stem cells [103]. The reason for these apparent discrepancies is still unknown but could be attributed to the different types of cell cultures and experimental procedures.

### 3.2. C/EBP

The CCAAT/enhancer-binding proteins (C/EBPs) encompass a family of six transcription factors with structural and functional homologies, but with different tissue specificity and transactivating ability.

C/EBPα was the first member cloned. Expression patterns of C/EBPα mRNA are similar in the mouse and human with measurable levels in liver, fat, intestine, lung, adrenal, peripheral blood mononuclear cells, and placenta. Similar to PPARγ, the role of C/EBPα as a mediator of BPA effects is currently under debate. Indeed, while Somm et al. have observed an increase of C/EBPα expression in adipocytes from BPA-exposed female rats [99], Atlas et al. did not find any effect of BPA on this transcription factor in 3T3-L1 cells [95]. Very recently, Salehpour et al. have described that BPA-induced triglyceride accumulation in human adipose-derived mesenchymal stem cells may be related not only to the upregulation of PPARγ and C/EBPα but also to the increase of C/EBPβ gene expression, suggesting that other members of the C/EBP family may be involved in the metabolic damage caused by BPA [105].

BPA exposure has been associated with liver dysfunction and diseases. De Benedictis et al. have shown that in fetal livers from female but not from male mice fed a diet supplemented with 25 mg BPA/kg, the level of C/EBPα, which is essential for hepatocyte maturation, is downregulated by 50% compared to the control animals. The authors conclude that in mice, BPA disrupts fetal liver maturation in a sex-specific manner and hypothesize that the decrease in C/EBPα may be responsible for the altered expression of albumin, alpha-fetoprotein, and glycogen synthase [106].

### 3.3. NRF2

Nuclear factor erythroid-2-related factor 2 (Nrf2) is a basic leucine zipper transcription factor that protects against oxidative damage by regulating the expression of antioxidant proteins [107]. Jiang et al. have shown that Nrf2^−/−^ mice suffer from severe pathological renal alterations after treatment with pristane, a saturated terpenoid alkane inducing autoimmune diseases in rodents. Accordingly, an increased Nrf2 level can improve these alterations and protect from lupus nephritis [108]. Interestingly, a study by Dong et al. indicates that oral BPA administration to lupus-prone MRL/lpr mice decreases Nrf2 expression in renal tissue exacerbating lupus nephritis [109]. Thus, Nrf2 seems to play a protective role in BPA-induced renal damage. However, unlike the kidneys, Nrf2 impairs liver function [110]. Indeed, in the liver of leptin-deficient mice, constitutive activation of Nrf2 downregulates Kelch-like ECH-associated protein 1 (Keap1), increasing lipid accumulation [111]. Similarly, BPA induces Nrf2 via Keap1 inactivation in a human hepatoma cell line [112]. The possible association between BPA and Nrf2 has been further analyzed by Shimpi et al., who indicate that BPA (25 μg/kg/day) administration to pregnant CD-1 mice induces Nrf2 expression and its recruitment to the Srebp-1c promoter causing hepatic lipid deposition [113].

### 3.4. HOX

Hox genes are a group of related genes that encode for transcription factors characterized by a well-conserved DNA sequence known as the homeobox, of which the term “Hox” was originally a contraction. Thirty-nine HOX genes which are located in four clusters (A–D) have been found in humans and rodents. Hox genes are expressed during embryogenesis and early development, where they act as master transcriptional regulators. In adults, they are mainly involved in the maintenance of the normal phenotype [114,115]. Among the HOX genes, HOXA9, HOXA10, HOXA11, and HOXA13 are expressed in the female reproductive system, while HOXB9 and HOXC6 are involved in the development of the mammary gland [116,117]. Their misregulation may have deleterious effects. In particular, HOXA10 is important for normal decidualization and pregnancy; deregulation of its expression has been associated with several pathological conditions, including ectopic pregnancy, PCOS, endometriosis, hydrosalpinx, and improper implantation [118]. Elevated levels of HOXA10 in the uterine stromal cells of female pups exposed in utero to BPA (0.5 mg/kg–1.0 mg/kg) may mediate the decidualization defects [118].

Similar to HOXA10, HOXB9 and HOXC6 also play a physiological role in mammary gland development and are also overexpressed in several tumors, including breast cancer [119,120,121]. Several authors indicate that BPA increases HOXB9 and HOXC6 expression both in cultured human breast cancer cells (MCF7) and in the mammary glands of ovariectomized rats (25 μg/kg), suggesting these transcription factors as mediators of BPA harmful effects in breast tumor [122,123].

### 3.5. HAND2

Heart- and neural crest derivatives-expressed protein 2 (HAND2) is a basic helix-loop-helix transcription factor, involved in the establishment of a proper implantation environment for pregnancy. Li et al. have demonstrated that chronic exposure of female mice to BPA (60 or 600 μg/kg/day) decreases HAND2 expression in uterine stroma, affecting embryo implantation and formation of the decidua during early phases of pregnancy [124]. HAND2 is also involved in the development of ventricular chambers and in cardiac morphogenesis [125]. Its overexpression is associated with an excessive proliferation of cardiac progenitor cells, leading to enlargement of the heart and increased size of the outflow tract [126]. Interestingly, emerging evidence indicates an association between cardiovascular diseases and BPA, also due to the presence of several BPA target receptors in the cardiac tissue [127]. In addition, more recent data suggest an important role of BPA in impairing the differentiation of some cardiac progenitors [128,129,130]. In particular, Lombò et al. have shown that more than 30% of zebrafish embryos exposed to BPA (4000 μg/L) display cardiac edema, defects in looping and ballooning, blood accumulation, and elongation of heart chambers. Moreover, BPA significantly increases ERβ expression and H3K9 and H4K12 histone acetylation which may be, in turn, responsible for HAND2 upregulation and the higher percentage of heart malformations [130]. Thus, HAND2 represents a key regulator of several organs, including the uterus and heart, and impairment of its expression by BPA through genetic and epigenetic mechanisms may cause reproductive and cardiac disorders.

## 4. BPA-Regulated Epigenetic Mechanisms

Epigenetic changes are mitotically heritable chemical modifications able to affect chromatin three-dimensional conformation and, consequently, gene expression. Environmental factors such as nutritional agents and xenobiotic contaminants modulate epigenetic patterns, influencing DNA methylation of the CpG dinucleotides, post-translational chemical modifications of histone tails, and small non-coding RNA levels.

The first evidence that BPA may modulate DNA methylation came in 2006 when neonatal exposure to a low environmentally relevant dose of BPA was shown to increase the susceptibility of rats to neoplastic prostatic lesions, inducing early and prolonged phosphodiesterase type 4 variant 4 (PDE4D4) gene hypomethylation and elevated expression [131]. Similarly, Dolinoy et al. found that prenatal exposure of Agouti mice to BPA leads to a shift in the coat color phenotype of genetically identical individuals of the offspring. This phenomenon is due to a BPA-induced reduction in methylation of nine CpG sites located in an intracisternal A particle retrotransposon upstream of the agouti gene [132]. Later, BPA epigenetic effects have been characterized by dose-response experiments supplementing the maternal diet with three different amounts of BPA and using the yellow agouti gene as epigenetic biosensor [133]. Further studies have evidenced the relevance of the prenatal window for BPA-induced epigenomic changes. Indeed, in CD-1 mice, BPA is able to decrease the methylation of the HOXA10 gene when intraperitoneally injected in utero, deregulating the programmed gene expression during development and affecting embryo viability. In contrast, adult mice exposed to the same amount of BPA did not modify the HOXA10 methylation pattern [134]. Methylation is not the only BPA-induced epigenetic modification. Several reports show the ability of BPA to specifically interfere with the expression of multiple microRNAs (miRNAs). This is not surprising since BPA is an estrogen mimic. In this regard, it may be involved in miRNA processing and in direct regulation of specific miRNAs owing to estrogen response elements (EREs) in their promoters [135]. The impact of in utero BPA exposure on histone chemical modifications has been explored too but in less detail. It is known that BPA prenatal exposure upregulates the expression of the histone methyltransferase Enhancer of Zeste Homolog 2 (EZH2), increasing trimethylation of histone 3 (H3) at lysine 27 (H3K27me3) in the mammary gland, which represents a marker of transcriptional activation typical of breast cancer cells [136]. Moreover, BPA exposure induces histone H3K4 trimethylation at the transcriptional initiation site of the alpha-lactalbumin gene, enhancing its expression [137].

### 4.1. BPA-Induced Epigenetic Modifications in Metabolism

The discovery that BPA is epigenetically toxic further encouraged the study of the etiology of several complex diseases, such as type 2 diabetes (T2D), obesity, and cancer. Indeed, several metabolic pathways are significantly modified by BPA action on the epigenome. In sheep fetal ovaries, the expression of miRNAs is altered by prenatal BPA, affecting insulin homeostasis and 15 of the differentially expressed miRNAs that are potentially involved in the regulation of genes related to insulin signaling [138]. Moreover, the utilization of genome-wide analysis together with in-depth quantitative site-specific CpG methylation has allowed one to discover and validate modifications in DNA methylation patterns in the BPA-treated mouse liver and to identify cancer- and metabolism-related pathways [139]. Ma et al. have observed that BPA administration to pregnant Wistar rats results in increased insulin resistance and reduced hepatic glycogen storage in the offspring. In parallel, DNA methyltransferase 3B mRNA is overexpressed, and hepatic global DNA methylation is decreased. In contrast, promoter hypermethylation of hepatic glucokinase (Gck) and a concomitant decreased gene expression of this enzyme has been noted, leading to diminished glycogen synthesis. These data support the key role of BPA-induced epigenetic changes in fetal reprogramming in the pathogenesis of metabolic disorders [140]. However, there are also contrasting data published by van Esterik et al. showing that perinatal BPA exposure does not influence DNA methylation in the liver. According to the authors, discordant results are probably due to species, strain, and tissue used, discrepancies in BPA dose and administration method, and background diet and/or other unidentified factors [141]. Anyway, BPA ability to impair glucose homeostasis and hepatic Gck promoter methylation in F2 offspring through maternal exposure has been also investigated in Sprague–Dawley (S–D) rats by Li et al. [142]. These authors show that BPA-treated F2 offspring feature significantly higher glucose intolerance and insulin resistance and decreased Gck protein and mRNA levels, compared to the untreated control. Accordingly, the impaired methylated status of Gck promoter in the liver of BPA-treated F2 offspring and in the sperm of F1 generation confirms that oral BPA administration during gestation and lactation worsens the risk of T2D and its progression in the F2 generation. Thus, BPA-induced epigenetic changes allow the transmission of alterations of glucose metabolism through generations [142]. Other evidence indicates that BPA-induced epigenetic changes may affect the expression of genes relevant for hepatic function. Indeed, upon 10 months exposure to BPA, CD-1 mice feature a decrease of DNA methyltransferase levels, accompanied by hypomethylation and overexpression of genes involved in lipid synthesis, such as Srebf1 and Srebf2. From the metabolic point of view, BPA-treated mice are characterized by obesity and by anomalies of glucose and lipid metabolism, such as increased fasting blood glucose and serum insulin and significant hepatic accumulation of triglycerides and cholesterol [143]. Triglycerides accumulation promotes the development of non-alcoholic fatty liver disease (NAFLD). Interestingly, male C57BL/6 mice exposed to BPA by oral gavage for 90 days display a NAFLD-like phenotype paralleled by reduced expression of miR-192, responsible for the upregulation of Srebf1 and, in turn, of several genes involved in de novo lipogenesis. In these mice, exposure to BPA impairs hepatic insulin signaling and induces systemic insulin resistance [144]. Moreover, in male S–D rats, early-life BPA exposure contributes to the development of NAFLD in adulthood and exacerbates the deleterious long-term effects of a post-weaning high fat diet. The proposed mechanism includes the induction of DNA hypermethylation within the Carnitine palmitoyltransferase 1a (Cpt1a) gene, encoding the enzyme regulating the transport of long-chain fatty acids into the mitochondria for β-oxidation [145]. Hypermethylation, in turn, leads to the down-regulation of Cpt1a expression and to a consequent accumulation of free fatty acids. Interestingly, BPA affects liver homeostasis through histones modifications, too. Indeed, it modifies several histone marks, including H3Me2K4, in histone tails within the Cpt1a gene, thus reducing the binding of several transcription factors to the Cpt1a gene. These alterations are paralleled by the reduction in expression levels of Kmt2c, which methylates and activates H3Me2K4 [146].

However, the BPA impact on lipid metabolism has been further clarified by studies performed on adipose tissue, which plays an undeniable key role in the pathogenesis of insulin resistance and T2D. Several studies performed in animal and cellular models highlighted BPA contribution to the development of obesity, probably due to its dose-related enhancing effect on adipocyte differentiation observed in 3T3-L1 cells. At the molecular level, BPA (80 μM) significantly increases global DNA methylation in 3T3-L1 cells [147]. Further studies performed in murine preadipocytes have pointed out the key role of miR-21-a-5p, whose levels are decreased by BPA exposure. It has been shown that that miR-21-a-5p interferes with MKK3/p38/MAPK, blocking BPA induced adipocytes differentiation. Interestingly, miR-21-a-5p overexpression mitigates BPA obesogenic effect in vivo [147]. In humans, prenatal BPA exposure seems to favor overweight phenotype in children, modifying methylation of CpG sites, including one hypo-methylated CpG in the promoter of the mesoderm-specific transcript gene (MEST) encoding an obesity-related member of the α/β hydrolase fold family [148].

Beside affecting insulin resistance and obesity, BPA action on DNA methylation deeply affects T2D pathogenesis and progression impacting on beta cells function. There is a lot of evidence that, when F0 pregnant S–D rats are exposed to BPA during gestation and lactation, sperm of adult F1 male rats and islets of male F2 offspring feature Igf2 hypermethylation and decreased expression. These molecular alterations in male F2 offspring are paralleled by impaired glucose tolerance and beta-cell dysfunction [149]. BPAs impact on specific miRNAs expression plays an emerging role in beta-cell dysfunction, too. In particular, miR-338 has been identified for its involvement in beta-cells response to BPA. Indeed, in vitro studies performed in primary islets treated for 48 h with BPA have revealed that islets are unable to compensate for long-term effects of BPA toxicity, featuring reduced glucose-stimulated insulin secretion and downregulation of Pdx1 expression. Interestingly, the authors show that Pdx1 serves as a target of miR-338 and that long-term BPA treatment upregulates miR-338 levels, determining a decrease in Pdx-1 expression and subsequently, a lack of compensatory insulin secretion [150]. Other studies have evidenced the ability of BPA to affect glucose metabolism modifying histone code. Indeed, maternal exposure to BPA significantly reduces pancreatic beta-cell mass and Pdx1 expression levels at birth and at gestational day 15.5. Decreased expression of Pdx1 is paralleled by histones H3 and H4 deacetylation, by demethylation H3K4 and by methylation of H3K9. These alterations of histone code at the promoter of Pdx1 lead to a compact chromatin structure and are conserved in adult life together with glucose intolerance [151].

### 4.2. BPA-Induced Epigenetic Modifications in Cancer

BPA induced epigenetic changes can make a decisive contribution to the pathogenesis of hormone-dependent cancer, such as breast and prostate cancer [152]. Dhimolea et al. have found that in utero, BPA exposure is accompanied by relevant transcriptional changes and genome-wide epigenetic modifications in the Wistar–Furth rat mammary gland from the end of exposure to adulthood [137]. In humans, BPA modifies the morphogenesis of the fetal mammary gland in females and induces gynecomastia in males [153]. Interestingly, low dose BPA exposure during the early stages of mammary gland development increases the risk of breast cancer in adult animals [154,155]. In vitro studies have shown that BPA exposure increases the proliferation of the human breast cancer cell line [51]. Some of the first evidence that BPA-induced epigenetic changes affect breast cancer pathogenesis was provided by the finding that treatment of MCF-7 cells with BPA increases mRNA and protein expression of EZH2, a histone methyltransferase linked to breast cancer risk. In parallel, histone H3 trimethylation is increased upon BPA incubation. Similar results about BPA induction of EZH2 expression have been obtained in mammary tissue of BPA-exposed mice [136,156]. Similarly, in MCF-7 cells and in mammary glands of S–D rats, expression of HOXC6, commonly upregulated in breast tumor tissue, increases upon BPA incubation by enhancing H3K4me3, histone acetylation and recruitment of RNA polymerase II [123]. Besides histones modifications, BPA-induced alterations of DNA methylation are involved in breast cancer pathogenesis. In human primary breast epithelial cells, low dose BPA leads to hypermethylation and silencing of lysosomal associated membrane protein 3 (LAMP3) gene [157], whose overexpression is usually linked to cancer invasiveness [158]. DNA methylation of BRCA-1 and p16 INK4 is also increased in human mammary epithelial cells treated with low dose BPA [159]. The procarcinogenic effect of BPA is supported by its ability to deregulate the expression of non-coding RNAs. BPA enhancing effect on the proliferation of MCF-7 cells is paralleled by the overexpression of oncogenic miR-21, miR-19a, and miR-19b [160] and by the silencing of miR-19 downstream targets, such as PTEN [161]. Moreover, in human placental cell lines treatment with BPA induces miR-146a upregulation, linked to the development of triple-negative breast cancer [162].

Prostate carcinogenesis is also affected by BPA exposure in both rats [163] and humans [164]. Genome-wide DNA methylation analysis in rodent models has shown that neonatal exposure to BPA induces permanent differential methylation in 86 genes, and increases susceptibility to prostate cancer [165]. At the molecular level, BPA increases prostate stem-progenitor cell self-renewal and upregulates the expression of genes connected to human prostate cancer in a dose-dependent manner. Dose-specific changes in the DNA methylation of genes such as Creb3L4, Tpd52, Pitx3, Paqr4, and Sox2 have been indeed observed upon postnatal BPA exposure in male neonatal S–D rats [166]. A whole-genome microarray performed in healthy primary human prostate epithelial cells has shown that BPA treatment affects the expression of genes relevant for cancer development and progression in prostate cells, involved in pathways modulating angiogenesis, cell proliferation, cell cycle, DNA replication and repair, metabolism, inflammation, and immune response pathways. In parallel, BPA deregulates the expression of transcripts relevant to epigenetic changes, such as histone and DNA methylation modifying enzymes [167]. Similar results have been obtained by Fatma Karaman et al. who performed PCR arrays in the human prostate adenocarcinoma PC-3 cell line to investigate the transcriptional profiling of chromatin-modifying enzymes and DNA methylation levels of tumor suppressor genes including p16, Cyclin D2, and Rassf1. In particular, chromatin immunoprecipitation experiments have evidenced a BPA-induced specific histone modification affecting chromatin accessibility of p16. Taken together, these results have pointed out the functional role of BPA-induced epigenetic signatures, suggesting that both DNA methylation and histone modifications play a functional role in carcinogenesis and could represent molecular biomarkers of BPA-induced prostate cancer progression [168].

## 5. Conclusions

Most of the world population is still widely exposed to BPA, due to its large use in the production of polycarbonate plastic and to its release into foods and beverages. It is nowadays quite clear that BPA is a major risk factor for endocrine, immune, and oncological diseases. Indeed, this chemical has been now included in the list of banned substances in several products, such as cosmetics or baby bottles. However, several contrasting results about the toxic effects of BPA have been described. Discrepancies in the results may be due to the use of a wide range of BPA concentrations as well as to the different experimental models [169]. Hence, the interpretation of the results of toxicological and epidemiological studies about the effects of BPA has been complicated by the use of non-oral routes of administration in many experimental conditions, different doses, absence of dose-response relationships, or small numbers of test animals. In parallel, many efforts have been performed in order to elucidate the molecular mechanisms through which this compound acts. The integration of the knowledge about the BPA molecular pathways with epidemiology could certainly improve the comprehension of the toxic effects of BPA on human health.

As summarized in this review article, a growing body of evidence indicates that BPA action is initiated through binding to relatively specific hormone receptors, including sex hormone receptors (ERs and ARs) and thyroid hormone receptors, thereby directly regulating gene expression. Nonetheless, rapid non-genomic actions may be mediated by the membrane-associated ERs and/or GPR30, which in turn may elicit signal transduction pathways, finally recruiting key transcription factors involved in growth and differentiation as well as in energy and nutrient metabolism. Most intriguingly, all the upstream pathways may contribute to stable and inheritable modifications, by regulating epigenetic enzymes, which may also sustain earlier exposure to BPA [170] Figure 2.

Other chemicals including bisphenol S (BPS) and bisphenol F (BPF) have been evaluated as an alternative to BPA, without reaching encouraging results [171,172]. For instance, very recent studies indicate that BPS is as effective as BPA in promoting certain types of breast cancer, and even more harmful to the reproductive system [173]. In more detail, BPS stimulates the proliferation of breast cancer cells modulating cyclin D and E levels through ER-dependent signaling. In parallel, BPS increases the expression of genes involved in cellular attachment, adhesion, and migration inducing epigenetic and transcriptional modifications [174]. Thus, BPS is likely worthy of the same legal restriction as BPA [173].

Therefore, to date, the best practice to reduce the harmful effects of BPA is still the precaution of limiting the consumption of plastic materials and promoting the use of BPA-free products.

## Figures and Tables

**Figure 1 ijms-21-05761-f001:**
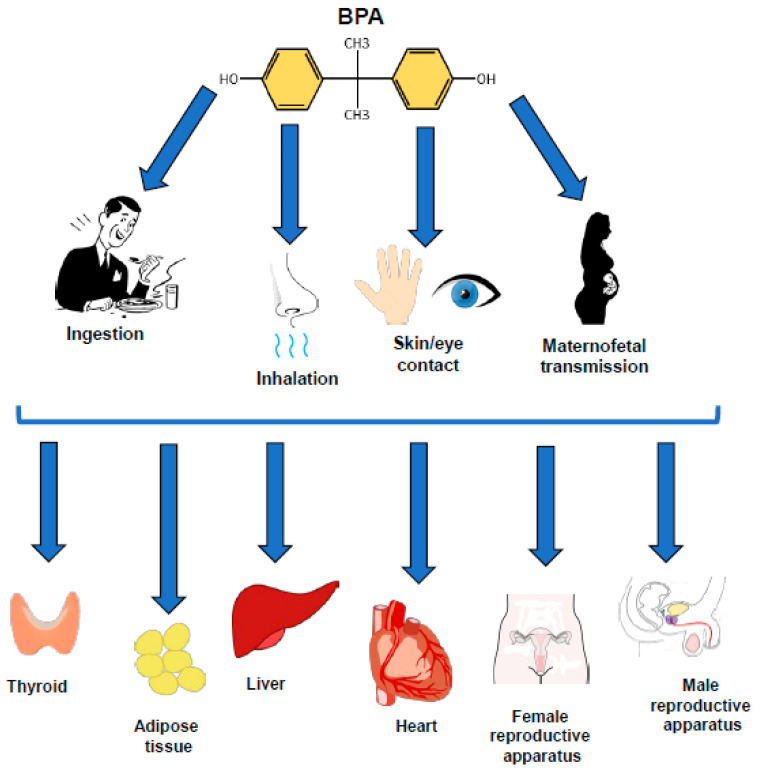
Potential BPA source and targets. BPA exposure sources include ingestion, maternofetal transmission, inhalation, skin, and eye contact. Once in the human body, BPA can negatively affect several targets, such as the thyroid, adipose tissue, liver, heart, female, and male reproductive apparatus. Images used to schematically represent anatomic parts and physiologic events were derived from openclipart.org and publicdomainvectors.org.

**Figure 2 ijms-21-05761-f002:**
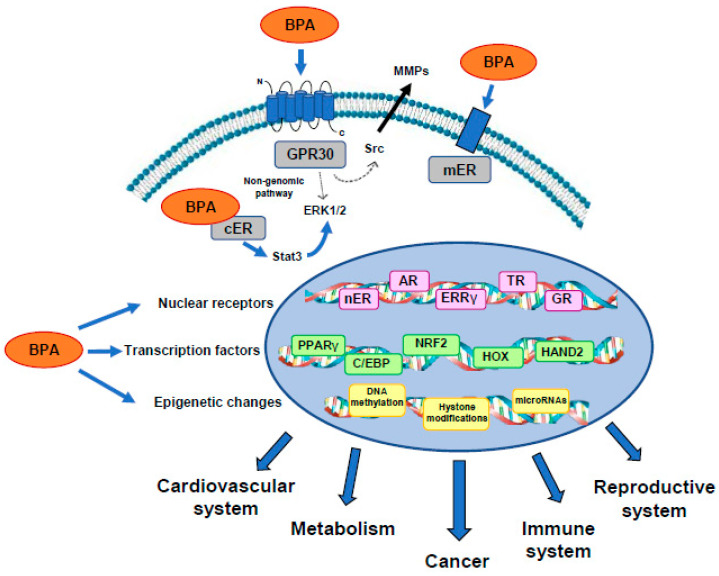
A potential integrative model of BPA molecular mechanisms. BPA exerts its deleterious effects on the cardiovascular system, metabolism, cancer, and immune and reproductive systems, by activating specific receptors, inducing transcription factors, and through epigenetic modifications.

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
