# Peer review of "Potential Mechanisms of Bisphenol A (BPA) Contributing to Human Disease"

_ijms, 2020, doi:10.3390/ijms21165761_

Round 1
Reviewer 1 Report
This manuscript sets out to describe potential mechanisms by which Bisphenol A (BPA) contributes to human disease. The authors describe several potential mechanisms by which BPA exposure may alter human biology, concluding that BPA can have widespread affects on human health. This manuscript is generally well-organized and attempts to answer an important question by reviewing the existing literature. However, the actual content of the review is lacking in data synthesis/interpretation, appropriate visualization, and does not consider/refute existing negative data related to BPA exposure.
Specific Comments:
Large number of grammatical errors throughout the manuscript. Please proofread and fix.
A significant amount of the information provided throughout the paper is not appropriately cited. For example, there are no citations for Section 1 (Introduction). Where is the support for the idea that persistent organic pollutants affect human health? Or for POPs showing up in various animal/human tissues? As another example, there are no citations for the first paragraph of section 2, despite presenting information sourced from previous studies. This is a repeated pattern that takes places several times throughout the paper, and is not appropriate for a review paper.
Figure 2 – What about the structure of BPA makes it lipophilic? The point being made in the text is accumulation in various tissues, but those tissues are not described in the figure. Instead, the authors simply write “health effects,” which is far too broad to provide an effective take-home message.
Figure 2 – The authors assert that microwaved plastic, bottles, and receipt paper are the “main sources” of BPA exposure. Presumably these are mostly related to ingestion? While these are indeed routes of BPA exposure, in the text, the authors mention a number of other potential sources, including inhalation, maternofetal, and skin/eye contact. So perhaps “potential sources” would be better terminology for Figure 2?
In Section 2 (the description of BPA), the authors make a number of mechanistic claims using supportive data from epidemiological studies. For example, in the description of BPA exposure in cashiers from receipt paper, the data presented is urine/blood levels from cashiers; this relationship is correlative and could be confounded by another variable. It should not be presented as causal. For these types of assertions, please provide direct evidence from dermal exposure studies. If no such studies are available, the language should be tempered.
Figure 3 – What are the “deleterious effects” described in the figure legend. This is not very descriptive; for this type of conceptual diagram, it would help to have the full picture shown. Furthermore, what does “Other receptors” refer to? This is once again broad language; why not just list them? If there are too many to list, this should probably be removed.
Figures 2 and 3 – I realize this is nitpicky, but please don’t use comic sans as a font in figures. It is cartoonish, and takes away from the idea that this is hard data being presented. Any Sans Serif font will look more professional.
In Sections 3, 4, and 5, the authors provide data to support the idea that BPA exposure can interact with receptor proteins, alter transcription factor expression, and affect epigenetic mechanisms. The data are compelling in their breadth, but the authors do little to interpret them. Instead, these sections read like simple lists of previous results. The authors need to connect these data into clear proposed pathways; these pathway could serve as useful summary figures to strengthen their arguments.
Strength of the authors' assertions, particularly in the conclusion, should be tempered. The authors do little to acknowledge or refute the large amount of negative BPA data that exists in the literature.
Author Response
-
- Large number of grammatical errors throughout the manuscript. Please proofread and fix.As requested, we have revised the entire manuscript and fixed grammatical errors.
- A significant amount of the information provided throughout the paper is not appropriately cited. For example, there are no citations for Section 1 (Introduction). Where is the support for the idea that persistent organic pollutants affect human health? Or for POPs showing up in various animal/human tissues? As another example, there are no citations for the first paragraph of section 2, despite presenting information sourced from previous studies. This is a repeated pattern that takes places several times throughout the paper, and is not appropriate for a review paper. We thank the Reviewer for raising this important point. We have modified the introduction, merging paragraph 1 and 2, revised the manuscript and added new references.
- Figure 2 – What about the structure of BPA makes it lipophilic? The point being made in the text is accumulation in various tissues, but those tissues are not described in the figure. Instead, the authors simply write “health effects,” which is far too broad to provide an effective take-home message. As suggested by the Reviewer, we have better described in the text the meaning of BPA lipophilicity (P3/L89). We have also shown in the new figure 1 the potential sources of BPA exposure and the tissues negatively affected by BPA.
- Figure 2 – The authors assert that microwaved plastic, bottles, and receipt paper are the “main sources” of BPA exposure. Presumably these are mostly related to ingestion? While these are indeed routes of BPA exposure, in the text, the authors mention a number of other potential sources, including inhalation, maternofetal, and skin/eye contact. So perhaps “potential sources” would be better terminology for Figure 2? Please see point 3.
- In Section 2 (the description of BPA), the authors make a number of mechanistic claims using supportive data from epidemiological studies. For example, in the description of BPA exposure in cashiers from receipt paper, the data presented is urine/blood levels from cashiers; this relationship is correlative and could be confounded by another variable. It should not be presented as causal. For these types of assertions, please provide direct evidence from dermal exposure studies. If no such studies are available, the language should be tempered. We thank the Reviewer for raising this important point. We have tempered the sentence as follows: “Moreover, very high plasma and urine levels of BPA have been found in the cashiers, the latter being more in contact with the thermal paper” (P3/L83).
- Figure 3 – What are the “deleterious effects” described in the figure legend. This is not very descriptive; for this type of conceptual diagram, it would help to have the full picture shown. Furthermore, what does “Other receptors” refer to? This is once again broad language; why not just list them? If there are too many to list, this should probably be removed. As suggested by the Reviewer, we have modified the old figure 3 describing more in the detail the molecular mechanisms of BPA action. We hope that the new figure 2 is now clearer.
- Figures 2 and 3 – I realize this is nitpicky, but please don’t use comic sans as a font in figures. It is cartoonish, and takes away from the idea that this is hard data being presented. Any Sans Serif font will look more professional. Done as requested. We have changed comic sans typeface with arial.
- In Sections 3, 4, and 5, the authors provide data to support the idea that BPA exposure can interact with receptor proteins, alter transcription factor expression, and affect epigenetic mechanisms. The data are compelling in their breadth, but the authors do little to interpret them. Instead, these sections read like simple lists of previous results. The authors need to connect these data into clear proposed pathways; these pathway could serve as useful summary figures to strengthen their arguments. We thank the reviewer for this important point. We have now attempted to interpret the literature reports in a unifying picture. This is explained in the Conclusion section (please see P15-L500) and in the novel Figure 2.
- Strength of the authors' assertions, particularly in the conclusion, should be tempered. The authors do little to acknowledge or refute the large amount of negative BPA data that exists in the literature. As requested by the reviewer, we have tempered the conclusion and sentence has been modified as follows: “It is nowadays quite clear that BPA is a major risk factor for endocrine, immune and oncological diseases. Indeed, this chemical has been now included in the list of banned substances in several products, such as cosmetics or baby bottles. However, several contrasting results about the toxicity effects of BPA have been described. Discrepancies in the results may be due to the use of a wide range of BPA concentrations as well as to the different experimental models” (P15/L501).
Reviewer 2 Report
IJMS-874935: Mechanisms of BPA Action
The authors have presented a clear review of the many mechanistic pathways by which bisphenol a (BPA) exerts its physiological effects, many of which are associated with negative health outcomes. In general, the paper is well-written and comprehensive.
My main requests for rewriting/editing are for the abstract and introductory sections. Once the authors get into the mechanistic sections, they are on stronger footing.
Abstract & Introduction para 1:
The description of BPA as being resistant to environmental degradation is incorrect. This is further complicated by the description in the first paragraph of the introduction of BPA as a persistent organic pollutant (POP). The term POP refers to chemicals whose half-lives are in years, more commonly decades and resistant to environmental degradation, often being lipophilic and bioaccumulating in wildlife and humans.
While BPA is ubiquitous in our environment and our bodies, this is more a reflection of its high-volume use and disposal rates, leading to high levels of constantly renewed contamination by the chemical. But its half-life in the environment is actually quite short compared to traditional POPs. One set of estimates, based on an extensive review of the literature, indicates the have life of BPA in the environment in days -- not years or decades as 38 for water, 340 for soil & sediment, and 0.2 for air.** And the half-life in humans is on the order of days, depending on exposure route, etc.
**Corrales et al. 2015. Global assessment of bisphenol A in the environment: Dose Response 13(3):1-29
The wording in both the abstract and introduction need to be revised.
Abstract:
It seems odd to say that it is "not well known how BPA exerts its deleterious effects” (emphasis mine) and then present a comprehensive 18 page review of the mechanisms by which we now understand BPA’s mechanistic properties. While certainly more has yet to be studied and revealed, this sentence should be removed.
It would be appropriate in this abstract of a review of BPA mechanisms to be a bit more informative about the content of the paper, in the least to mention the specific receptor systems you will be discussing. In other words, expand briefly on the second to last sentence.
This is an extensive and comprehensive review. But I think the words “exhaustive analysis” are inappropriate.
- Bisphenol A:
Para 1: The authors do mention that BPA’s estrogenic properties have been known since 1936 (described in introduction to section on BPA interactions with specific receptors), but it would be appropriate to include this information upfront in the brief history of the compound at the bottom of page 2.
Similarly, although the authors mention in the final paragraph on page 4 some of the possible negative health impacts of exposures to BPA, it would be appropriate to move that information to the introductory comments at the bottom of page 2, after “compromising their physiological function”.
Figures:
I don’t think that the three figures add much to the introduction. They are all very basic and do not add, or simplify the text. In the cases of Figs. 2 & 3, they directly repeat, in simple cartoon, quite straight-forward information offered in the text.
On the other hand, having the chemical structure of BPA along with other figures that present the much more complex mechanistic information in the body of the text, would be quite helpful. Figures that picture the pathways for each of the categories of pathways described, and perhaps even a very complicated integrative model, would make the very complex text information accessible in another way.
Author Response
- Abstract & Introduction para 1: The description of BPA as being resistant to environmental degradation is incorrect. This is further complicated by the description in the first paragraph of the introduction of BPA as a persistent organic pollutant (POP). The term POP refers to chemicals whose half-lives are in years, more commonly decades and resistant to environmental degradation, often being lipophilic and bioaccumulating in wildlife and humans. While BPA is ubiquitous in our environment and our bodies, this is more a reflection of its high-volume use and disposal rates, leading to high levels of constantly renewed contamination by the chemical. But its half-life in the environment is actually quite short compared to traditional POPs. One set of estimates, based on an extensive review of the literature, indicates the have life of BPA in the environment in days -- not years or decades as 38 for water, 340 for soil & sediment, and 0.2 for air.** And the half-life in humans is on the order of days, depending on exposure route, etc. We thank the reviewer for raising this important point. We have clarified this point and changed the sentence in the introduction as follows: “Thus, the inclusion of BPA in POPs category is controversial. Indeed, although not technically a persistent organic pollutant because of its short half-life, it is often grouped together with other POPs as it can accumulate in several human tissues and organs, contributing to the pathogenesis of several diseases” (P2/L62).
We have also added the suggested reference in the text: Corrales et al. 2015. Global assessment of bisphenol A in the environment: Dose Response 13(3):1-29 (P3/65).
- The wording in both the abstract and introduction need to be revised.
Done as requested.
- Abstract: It seems odd to say that it is "not well known how BPA exerts its deleterious effects” (emphasis mine) and then present a comprehensive 18 page review of the mechanisms by which we now understand BPA’s mechanistic properties. While certainly more has yet to be studied and revealed, this sentence should be removed. It would be appropriate in this abstract of a review of BPA mechanisms to be a bit more informative about the content of the paper, in the least to mention the specific receptor systems you will be discussing. In other words, expand briefly on the second to last sentence. Thanks for this observation. We have rewrote the abstract taking into account aforementioned suggestions. We hope that now it is clearer
- This is an extensive and comprehensive review. But I think the words “exhaustive analysis” are inappropriate. In the new abstract “exhaustive analysis” has been changed with “extensive and comprehensive analysis” (P2/L40).
- Bisphenol A: Para 1: The authors do mention that BPA’s estrogenic properties have been known since 1936 (described in introduction to section on BPA interactions with specific receptors), but it would be appropriate to include this information upfront in the brief history of the compound at the bottom of page 2. As requested by the reviewer, we have merged the introduction paragraphs 1 and 2 and changed the sentence as follows: ”The first evidence about the action mechanisms of BPA was obtained in 1936 by Dowds and Lawson who discovered its estrogenic properties in vivo. In 1997 the involvement of estrogen receptors, ER α and β, in BPA action was described, while later, other mechanisms emerged” (P3/L65).
“Because of its lipophilic nature (logP of 3.4), BPA has the ability to accumulate in different tissues, compromising their physiological functions.” (P3/L89).
- Similarly, although the authors mention in the final paragraph on page 4 some of the possible negative health impacts of exposures to BPA, it would be appropriate to move that information to the introductory comments at the bottom of page 2, after “compromising their physiological function”. Please see point 5.
- Figures: I don’t think that the three figures add much to the introduction. They are all very basic and do not add, or simplify the text. In the cases of Figs. 2 & 3, they directly repeat, in simple cartoon, quite straight-forward information offered in the text. On the other hand, having the chemical structure of BPA along with other figures that present the much more complex mechanistic information in the body of the text, would be quite helpful. Figures that picture the pathways for each of the categories of pathways described, and perhaps even a very complicated integrative model, would make the very complex text information accessible in another way. As suggested by the Reviewer, we have modified the old figure 2 describing the potential sources of BPA exposure and the tissues where BPA acts. Please see new figure 1. Similarly, we have modified the old figure 3 describing the molecular mechanisms of BPA action. We hope that the new figures 1 and 2 are now more clear.
Reviewer 3 Report
This manuscript by. Cimmino et al., provides an analysis of the molecular action of bisphenol A
The manuscript is interesting
The introduction is very short, I recommend improving the introduction
I suggest the authors to add these recent manuscripts regarding the bisphenol a
Rotondo E, Chiarelli F. Endocrine-Disrupting Chemicals and Insulin Resistance
in Children. Biomedicines. 2020 May 28;8(6):E137.
Abraham A, Chakraborty P. A review on sources and health impacts of bisphenol
- Rev Environ Health. 2020 Jun 25;35(2):201-210
Castellini C, Totaro M, Parisi A, D'Andrea S, Lucente L, Cordeschi G,
Francavilla S, Francavilla F, Barbonetti A. Bisphenol A and Male Fertility:
Myths and Realities. Front Endocrinol (Lausanne). 2020 Jun 12;11:353.
Wehbe Z, Nasser SA, El-Yazbi A, Nasreddine S, Eid AH. Estrogen and Bisphenol
A in Hypertension. Curr Hypertens Rep. 2020 Feb 29;22(3):23
Thoene M, Dzika E, Gonkowski S, Wojtkiewicz J. Bisphenol S in Food Causes
Hormonal and Obesogenic Effects Comparable to or Worse than Bisphenol A: A
Literature Review. Nutrients. 2020 Feb 19;12(2):532.
Jung N, Maguer-Satta V, Guyot B. Early Steps of Mammary Stem Cell
Transformation by Exogenous Signals; Effects of Bisphenol Endocrine Disrupting
Chemicals and Bone Morphogenetic Proteins. Cancers (Basel). 2019 Sep
12;11(9):1351.
Author Response
- The introduction is very short, I recommend improving the introduction. I suggest the authors to add these recent manuscripts regarding the bisphenol a
- Rotondo E, Chiarelli F. Endocrine-Disrupting Chemicals and Insulin Resistance in Children. Biomedicines. 2020 May 28;8(6):E137.
- Abraham A, Chakraborty P. A review on sources and health impacts of bisphenol Rev Environ Health. 2020 Jun 25;35(2):201-210
- Castellini C, Totaro M, Parisi A, D'Andrea S, Lucente L, Cordeschi G, Francavilla S, Francavilla F, Barbonetti A. Bisphenol A and Male Fertility: Myths and Realities. Front Endocrinol (Lausanne). 2020 Jun 12;11:353.
- Wehbe Z, Nasser SA, El-Yazbi A, Nasreddine S, Eid AH. Estrogen and Bisphenol A in Hypertension. Curr Hypertens Rep. 2020 Feb 29;22(3):23
- Thoene M, Dzika E, Gonkowski S, Wojtkiewicz J. Bisphenol S in Food Causes Hormonal and Obesogenic Effects Comparable to or Worse than Bisphenol A: A Literature Review. Nutrients. 2020 Feb 19;12(2):532.
- Jung N, Maguer-Satta V, Guyot B. Early Steps of Mammary Stem Cell Transformation by Exogenous Signals; Effects of Bisphenol Endocrine Disrupting Chemicals and Bone Morphogenetic Proteins. Cancers (Basel). 2019 Sep 12;11(9):1351.
As indicated by the reviewer, we have rewrote the abstract and added the suggested references in the text.
Round 2
Reviewer 1 Report
Thank you for putting in so much effort to respond to initial reviews. The manuscripts is vastly improved as a result of the changes. In particular, the new figures look great.
While the grammar/spelling is quite improved, there are still some grammatical errors throughout the manuscript. For example, in the abstract, the sentence "BPA can contaminate food, beverage, air and soil, then accumulating in several human tissues and organs and resulting harmful to human health through different molecular mechanisms." should read "BPA can contaminate food, beverage, air and soil, accumulating in several human tissues and organs and resulting in harm to human health through different molecular mechanisms."
As another example, in the first sentence of the introduction, "Persistent organic pollutants (POPs) are organic compounds able to resist to most of the degradation processes and to bioaccumulate in the environment, affecting human health" should read "Persistent organic pollutants (POPs) are organic compounds able to resist most degradation processes and bioaccumulate in the environment, affecting human health." I recognize that these are subtle changes, but they are important for convincing the reader that the authors have the necessary knowledge base to be writing a review article.
These types of minor errors occur throughout the manuscript. Please proofread again and fix.
In addition, I noticed that while the authors now include citations for their description of BPA -- "Bisphenol A (BPA) is an organic synthetic compound of 288 Da, having the chemical formula (CH3)2C(C6H4OH)2 and belonging to the group of diphenylmethane derivatives and bisphenols, with two hydroxyphenyl groups" -- this text is nearly identical to the definition from wikipedia (see "Bisphenol A" entry https://en.wikipedia.org/wiki/Bisphenol_A). To avoid potential claims of plagiarism and unwanted retractions, please restate this information in a different manner.
LINES 97-112: While this added information is appreciated, the data presented are unclear as written. From what models do these results come from? Are they all from human studies? Mice? Zebrafish? Please indicate what model/cells/etc. were used for each study. Otherwise, these data could give a false impression that all of these results are seen across the board, in ALL studied models. Also, it seems that this would be a good time to again present the fact that many of these biological effects of BPA exposure remain relatively controversial. For example, somewhere near LINE 95, you could introduce the fact that different models/doses have led to different effects, leading to some uncertainty regarding BPA's deleterious effects. Then you can try to push back against the idea that it doesn't cause negative health effects using the data presented in LINES 97-112.
Author Response
- While the grammar/spelling is quite improved, there are still some grammatical errors throughout the manuscript. For example, in the abstract, the sentence "BPA can contaminate food, beverage, air and soil, then accumulating in several human tissues and organs and resulting harmful to human health through different molecular mechanisms." should read "BPA can contaminate food, beverage, air and soil, accumulating in several human tissues and organs and resulting in harm to human health through different molecular mechanisms." Done as requested
- As another example, in the first sentence of the introduction, "Persistent organic pollutants (POPs) are organic compounds able to resist to most of the degradation processes and to bioaccumulate in the environment, affecting human health" should read "Persistent organic pollutants (POPs) are organic compounds able to resist most degradation processes and bioaccumulate in the environment, affecting human health." I recognize that these are subtle changes, but they are important for convincing the reader that the authors have the necessary knowledge base to be writing a review article. Done as requested.
- These types of minor errors occur throughout the manuscript.
Please proofread again and fix. In addition, I noticed that while the authors now include citations for their description of BPA -- "Bisphenol A (BPA) is an organic synthetic compound of 228 Da, having the chemical formula (CH3)2C(C6H4OH)2 and belonging to the group of diphenylmethane derivatives and bisphenols, with two hydroxyphenyl groups" -- this text is nearly identical to the definition from wikipedia (see "Bisphenol A" entry https://en.wikipedia.org/wiki/Bisphenol_A). To avoid potential claims of plagiarism and unwanted retractions, please restate this information in a different manner. We thank the Reviewer for raising this important point and we apologize for this mistake. We have changed the sentence as follows: “Bisphenol A (BPA) is an organic synthetic compound with a molecular weight of 228 Da and the chemical formula (CH3)2C(C6H4OH)2. It is included in the group of diphenylmethane derivatives and bisphenols, with two hydroxyphenyl groups.”
- LINES 97-112: While this added information is appreciated, the data presented are unclear as written. From what models do these results come from? Are they all from human studies? Mice? Zebrafish? Please indicate what model/cells/etc. were used for each study. Otherwise, these data could give a false impression that all of these results are seen across the board, in ALL studied models. As requested, we have modified the text and better specified where the studies have been performed (P5/L100).
Also, it seems that this would be a good time to again present the fact that many of these biological effects of BPA exposure remain relatively controversial. For example, somewhere near LINE 95, you could introduce the fact that different models/doses have led to different effects, leading to some uncertainty regarding BPA's deleterious effects. Then you can try to push back against the idea that it doesn't cause negative health effects using the data presented in LINES 97-112. As requested, we have modified the text indicating the existence of controversial studies about BPA effect (P5/L98).